# Application of Potassium after Waterlogging Improves Quality and Productivity of Soybean Seeds

**DOI:** 10.3390/life12111816

**Published:** 2022-11-07

**Authors:** Muhammad Abdullah Al Mamun, Umakanta Sarker, Muhammad Abdul Mannan, Mohammad Mizanur Rahman, Md. Abdul Karim, Sezai Ercisli, Romina Alina Marc, Kirill S. Golokhvast

**Affiliations:** 1Department of Agronomy, Faculty of Agriculture, Bangabandhu Sheikh Mujibur Rahman Agricultural University, Gazipur 1706, Bangladesh; 2Department of Genetics and Plant Breeding, Faculty of Agriculture, Bangabandhu Sheikh Mujibur Rahman Agricultural University, Gazipur 1706, Bangladesh; 3Department of Soil Science, Faculty of Agriculture, Bangabandhu Sheikh Mujibur Rahman Agricultural University, Gazipur 1706, Bangladesh; 4Department of Horticulture, Faculty of Agriculture, Ataturk University, 25240 Erzurum, Turkey; 5Food Engineering Department, Faculty of Food Science and Technology, University of Agricultural Sciences and Veterinary Medicine, 400372 Cluj-Napoca, Romania; 6Siberian Federal Scientific Center of Agrobiotechnology RAS, 2b Centralnaya street, Krasnoobsk 630501, Russia

**Keywords:** potassium, soybean, water logging, yield

## Abstract

Potassium (K) improves the stress tolerance of crop plants, which varies on the timing of K application and crop varieties. Soybean is a promising crop that can easily fit with the cropping pattern during kharif I season, when water logging occurs due to sudden rain. Therefore, an experiment was conducted to determine the effect of K management on the productivity and seed quality of soybean under normal and waterlogged conditions. The treatments comprised three factors, namely soybean genotypes (BU Soybean-1 and BU Soybean-2), waterlogging (WL) (control and WL for 4 days at the flowering stage (FS)), and K application (full dose as basal and 50% as basal +50% as top dress after termination of the flooding). The trial was laid out in a randomized complete block design with three replications. Findings revealed that BU Soybean-1 produced a higher number of pods and seeds pod^−1^ under control conditions with basal application of K. On the other hand, BU Soybean-2 produced taller plants and heavier grain, improving grain and straw yield under WL conditions when K was top dressed. The varieties absorbed a higher amount of nitrogen, phosphorus, and potassium under control conditions compared to WL when K was top dressed. Similarly, the seed protein content of both varieties was higher in the control condition with a top dressing of K. However, a higher percentage of seed germination was obtained from BU Soybean-2 in the control condition with a top dressing of K. Further, more electrical conductivity and more mean germination time were recorded in the case of BU Soybean-2 under WL with the basal application of K. Split application of 50% of recommended K fertilizer after the recession of flood water could be suggested for improved grain yield in flood-affected soybean growing areas.

## 1. Introduction

Soybean (*Glycine max* L.) has been cultivated since the early 1970s in Bangladesh, when the Mennonite Central Committee worked in the district of greater Noakhali. Recently, the cultivation of soybean has been extended dramatically from only 5000 ha in 2005 [1] to 57,670.85 ha in 2020–2021 in Bangladesh [2]. In Bangladesh, consciousness about the high protein and nutrient content of soybean is increasing day by day [3,4]. Plants are natural sources of biochemicals with numerous phenolics, antioxidants, vitamins, flavonoids, minerals, numerous pigments, dietary fiber, protein, and carbohydrates [5,6,7,8,9,10] for human health benefits. One hundred grams of dry seed of soybean contains 30–50% protein, 277 mg calcium, 15.7 mg iron, 280 mg magnesium, 704 mg phosphorus, 1797 mg potassium and 375 µg folic acid. Diversified adaptation strategies and nutritional value make soybean more popular to growers worldwide. However, many soybean-growing countries have encouraged farmers to produce more food through supporting soybean cultivation due to its multiple use and positive impact on soil [11]. Moreover, if we compare the soybean yield, we find that average yield is 1.8 t ha^−1^ in Bangladesh, while the global average is 2.76 t ha^−1^ [12,13].

In Bangladesh, the optimum sowing time for soybean is mid-December to mid-January, and the crop is harvested during April [14]. Planting soybean in January suffers from waterlogging (WL) at the pod formation stage in March due to a change in rainfall pattern. For example, heavy torrential rains in April 2017 and super cyclone Amphan in 2020 damaged soybean crops at the late pod development stage (physiological maturity) in the greater Noakhali and Bhola areas. In 2016, there was a cyclone called NADA in early November that caused heavy rains and delayed soybean sowing, which was supposed to be occur directly after harvesting Aman rice.

Soybean prefers adequate soil oxygen for maximum productivity, but WL reduces the amount of oxygen available to the plant. The WL condition is critical water stress, which affects the adaptation of soybean and reduces grain yield because it induces a significant detrimental effect on morphological and biochemical attributes of soybean. Different plant processes such as photosynthesis, accumulation of dry matter, plant growth, and formation of flowers and pods are marked as disturbed under the WL conditions [15,16,17]. Any abiotic stress such as WL/drought/salinity reduces the production of crops [18], by creating oxidative damage [19,20] by reactive oxygen species (ROS) [21], which eventually generate change, i.e., membrane, DNA, and protein damage, nutrient imbalance [22,23], and diminution in photosynthetic rates and changes color pigments [24,25,26]. To mitigate ROS, the plant has enhanced both enzymatic and non-enzymatic antioxidants, such as tocopherols betalain, ascorbic acids, carotenoids, betacyanin, betaxanthin, chlorophyll *a*, chlorophyll *b*, beta-carotene, phenolic and flavonoids [27,28,29,30,31,32], which detoxify the ROS. Under the WL condition, the plant suffers from a deficiency of oxygen, carbon dioxide, and light. Nitrogen (N) fixation is greatly affected by WL since the nodules of soybean fix N in the soil [33,34]. Under the WL condition, plants show different symptoms such as yellowing of leaves due to chlorosis, cell damage due to necrosis, and defoliation. However, the height of the plant is drastically reduced and N fixation is retarded under excess soil moisture conditions. As a result, 20–39% yield of soybean is reduced when soybean plants are exposed to flood at the R5 stage [35]. Soil WL caused an 18% yield loss in soybean around the world when flooded during the late vegetative stage [36]. Similarly, the grain yield of soybean decreased at a higher rate when it was exposed to WL in the R4 stage as compared to the R1 stage [37]. In the case of WL at the V2 and V3 growth stage, yield loss was 20% as reported by Sullivan et al. [38]. The yield loss in soybean was due to severe disease incidence, hypoxia, stunting of shoot height, reduced root and nodule formation under WL conditions. For better crop establishment, judicial application of input is essential [39]. Potassium (K) is one of the most important macronutrients for crop growth and development. This nutrient element has a great role in cereal and legume crops. It is used to uptake water and maintain cell turgidity. The formation and translocation of starch are also regulated by K within the plant. Translocation of nutrient and protein synthesis are also influenced by K. It helps the soybean plants to cope with different stresses, diseases, pests, and balanced uptake of other nutrients. It also helps in enzyme activation during nodulation [40] and has a prominent functionality in N and P uptake. Potassium enhances the photosynthetic rate and carbohydrate production, translocation, and metabolism. Therefore, it ultimately improves grain quality and yield. The root activity of plants is reduced under excess soil moisture conditions due to the low amount of K [41]. However, the demand for K in the plant is closely related to internal metabolic paths or the rate of phosphate recycling [42].

The K element has a viable function in the development, growth, and production processes of the plant [43], as it imparts its role in the morphologic and physiological characteristics of living plant cells [44], rather than only being part of the plant structure [45] Thus, proper time of K application can protect soybean crops from nutrient deficiency and can help recover from lodging and flood damage. Under moisture stress, K improves the stress tolerance [46] and increases the dry matter [47] and yield. Therefore, it is imperative to ensure the proper time of K application for sustainable agricultural production in K-deficient soil. Thus, the present experiment was undertaken to determine the effect of K management on the yield and seed quality of soybean under excess soil moisture conditions.

## 2. Materials and Methods

### 2.1. Location

A field experiment was conducted at the agronomy research field of Bangabandhu Sheikh Mujibur Rahman Agricultural University Gazipur, Bangladesh, during the rabi season 2020. The experimental site was located between 24°09′ N and 90°26′ E under the sub-tropical climatic zone with an elevation of 8.4 m from sea level. The textural class was silty clay, containing 40% clay, 45% silt, and 15% sand, having a pH of 6.1, soil organic matter 1.20%, total N 0.11%, available P 7.21 ppm, exchangeable K 0.19 meq/100 g soil, and available S of 11 ppm. The climate is sub-tropical in nature, characterized by moderately low temperatures associated with scanty rainfall during winter. The air temperature is low in the early crop growth stage and increased gradually from January to June. Total monthly precipitation is minimum up to April but dramatically increases from May (Table 1). Prior to experimentation, a soil sample was collected from the field to determine the physicochemical properties of the soil. The soil belongs to the Salna series under the Shallow Red Brown Terrace of the Argo-ecological zone (AEZ) Madhupur Tract (AEZ-28).

### 2.2. Land Preparation and Layout

The land was prepared very well by deep and cross plowing with tractor-drawn disc-plow and rotavator followed by laddering. All uprooted weeds and stubbles were incorporated into the soil. After one week, the plots were prepared as per design. Before layout preparation, the land was fertilized with urea, triple superphosphate, muriate of potash, gypsum, and zinc sulfate at 60, 170, 100, 100, and 10 kg ha^−1^, respectively. The fertilizers were uniformly incorporated into the plot before sowing seeds. However, muriate of potash was applied as per treatment. The unit plot size was 3 m × 4 m. Ridges were made around each plot to restrict the lateral movement of water. The blocks and unit plots were separated by 1.0 m and 1.5 m spacing, respectively.

### 2.3. Experimental Treatments and Design

The treatments comprised three factors, Factor A (soybean varieties): (i) BU Soybean-1 and (ii) BU Soybean-2, Factor B (WL): (i) control and (ii) WL for 4 days at the flowering stage), and Factor C (K application): (i) full dose as basal and (ii) 50% as basal +50% as top dress after the termination of the WL. The experiment was laid out in a randomized complete block design with three replications.

### 2.4. Seed Sowing and Crop Culture

Seeds of soybean varieties under this experiment were collected from the Department of Agronomy, Bangabandhu Sheikh Mujibur Rahman Agricultural University, Gazipur Bangladesh. The germination of the collected seed was 95%, which was confirmed by a germination test before sowing in the main field. Sowing was performed manually in lines maintaining the spacing of 30 cm from line to line and 5 cm from plant to plant. Immediately after sowing seeds, the plots were lightly irrigated to ensure uniform emergence. The seedlings emerged within five days after sowing (DAS). Thinning was performed during the appearance of the first trifoliate leaf stage, keeping one uniform and healthy seedling after every 5 cm distance in each row. Weeding and mulching were performed to keep the crop free from weeds. A sufficient amount of water was applied in each plot by supplemental irrigation twice per week up to the flowering stage of the crop.

### 2.5. Imposition of WL Stress

WL plots were surrounded by 30 cm deep polythene anchored into the ground and extending 30 cm above the ground to hold water. The WL treatment was imposed at the flowering stage (60 DAS). The WL stress was induced by flooding the plots completely up to 5 cm above the ground level. The treatments were continued up to 4 days of WL. Afterward, water was drained out from the treated plots. In the control (non-stress) treatment, water was applied twice per week.

### 2.6. Harvesting and Sampling Crops

Harvesting was performed at physiological maturity of the crop (turned brownish and became hard). A total of five plants were considered as a sample of those respective varieties for recording yield contributing characters. At each sampling, five plants were randomly selected from a single row. To avoid the border effect, the first and last rows of the plot were discarded during sampling. The sample plants were collected randomly. For taking yield data, a 1.8 m^2^ area was harvested, and seeds were threshed and dried. The grain weight was taken and adjusted at 14% moisture content. The plant stems were dried and straw was recorded.

### 2.7. Quantification of Yield Data

Plant height was measured by a meter scale of 100 cm. The plants were cut from the ground level. The height of five plants was measured from the base to the tip of the main shoot, and the height of the five plants was averaged. All pods from the five sample plants were counted, and the average value was taken. The pod having at least one seed was counted as a filled pod and the pod having no seed was counted as an unfilled pod. Pod length was measured on a small scale of 30 cm. After collecting all the pods, ten pods were selected randomly, the length was measured, and the mean value was recorded. After separating the seeds from the pods, they were counted by hand. Then, the average value was recorded. Weight of 100 seeds was recorded for each variety treatment-wise. Total seeds from a 1.8 m^2^ area were weighed by an electrical balance. The weight of seeds was converted to t ha^−1^. The moisture content of seeds was measured using a digital moisture meter and adjusted to 14% moisture using formula (1).
(1)$$\mathrm{Adjusted}\;\mathrm{weight}=\frac{W\times \left(100-M_{1}\right)}{\left(100-M_{2}\right)}\;✕\;100$$
where W is the fresh weight, and M_1_ and M_2_ are the fresh and adjusted moisture percent of the grain, respectively.

For taking straw yield, a 1.8 m^2^ area was harvested, and seeds were separated. The plant stems were dried, and straw was recorded and converted to t ha^−1^. Harvest index (HI) was determined using the following Formula (2):(2)$$\mathrm{HI}=\frac{\mathrm{Grain}\;\mathrm{yield}}{\mathrm{Grain}\;\mathrm{yield}\;+\;\mathrm{Straw}\;\mathrm{yield}}$$

### 2.8. Determination of Seed Quality Data

Germination of seed is the most important criterion of seed quality. One hundred seeds harvested from different treatments were used and replicated three times. Seeds were placed in a 9 cm petri dish containing filter paper soaked with distilled water. The petri-dishes were placed in an incubator at 30 °C until the completion of germination. Seedlings were counted every day, and a seed was considered to be germinated as the seed coat ruptured and the radical came out 2 mm in length. The final germination count was made according to ISTA [48]. Germination percentages were calculated by using the following Formula (3):(3)$$\mathrm{Germination}\;(\%)=\frac{\mathrm{No}.\;\mathrm{of}\;\mathrm{seeds}\;\mathrm{germinated}\;}{\mathrm{No}.\;\mathrm{of}\;\mathrm{seeds}\;\mathrm{incubated}\;\mathrm{for}\;\mathrm{germination}}\;✕\;100$$

The simplest method is to make preliminary germination counts at a standard time before germination is completed. The seed sample that produces the largest number of germinated seeds at the preliminary count will produce the fastest growing seedlings and the fastest stand establishment. The speed of germination of the seed sample was monitored by counting the germinated seedling at an interval of 24 h and counting until germination was completed. An index of the speed of germination was then calculated by adding the quotients of the daily counts divided by the number of days of germination. Thereafter, a germination index (GI) was computed by using the following Formula (4) to know the seed vigor [49].
(4)$$\mathrm{GI}=\frac{n\;}{d}$$
where n = number of seedlings emerging on the day ‘d’, d = day after planting

Seed vigor index (SVI) was calculated by using the following Formula (5):(5)$$\mathrm{SVI}=\frac{\mathrm{Seedling}\;\mathrm{length}\;\left(\mathrm{cm}\right)\;\mathrm{Germination}\;\left(\%\right)\;}{100}$$

Mean germination time (MGT) was calculated by the formula (6):(6)$$\mathrm{MGT}=\frac{n1\times \;d1+\;n2\times \;d2+\;n3\times \;d3+--------}{\mathrm{Total}\;\mathrm{number}\;\mathrm{of}\;\mathrm{days}}$$
where n = number of germinated seeds, d = number of days

### 2.9. Determination of Seed Coat Leakage in Seeds

The electrical conductivity (EC) of the soybean seeds was tested using the standard procedure to determine the quality of the seeds. The seeds were weighed on an analytical balance, immersed in 75 mL of deionized water in plastic cups, and kept in a germination chamber at 25 °C for 24 h. After the seed-soaking period, the electrical conductivity of the soaking solutions was determined in a conductivity meter. The results obtained were divided by the mass of fifty seeds and expressed in μS cm^−1^ g^−1^ of seeds [50].

### 2.10. Determination of Nutrient Composition in Seeds

The soybean seeds were dried at 70 °C for 72 h and ground by a Wiley Mill. The ground sample was digested in concentrated H_2_PO_4_, and the total N concentration was determined by the micro Kjeldahl method [51]. The concentration of P and K was analyzed by digesting a 0.2 g ground sample with 6 mL of 5:2 HNO_3_:HClO_4_ [51]. Total nutrient uptake was determined by the following formula (7):(7)$$\mathrm{Nutrient}\;\mathrm{in}\;\mathrm{grain}\;(\mathrm{kg}\;{\mathrm{ha}}^{-1})=\frac{\mathrm{Nutrient}\;\mathrm{in}\;\mathrm{grain}\;\left(\%\right)\;\mathrm{Grain}\;\mathrm{yield}\;\left(\mathrm{kg}\;{\mathrm{ha}}^{-1}\right)}{100}$$

The amount of protein present in seed samples was calculated from the N concentration of the seeds following formula (8).
Protein (%) = N (%) × 5.71(8)

### 2.11. Statistical Analysis

Replication-wise means data were obtained by averaging the sample mean [52]. All data were expressed as mean ± standard deviation of the triplicate measurements [53]. The collected data of different parameters were compiled and subjected to analysis of variance by using CropStat 7.2 statistical package program. The treatment means were compared using the DMRT at a 5% level of significance [54].

## 3. Results

### 3.1. Plant Height and Pod Production

There was no significant difference between control and WL treatment regarding plant height of soybean when K was applied either basally or top dressed (Table 2). However, BU Soybean-2 produced a taller plant under WL condition (48.86 cm) and BU Soybean-1 gave the shorter plant under control condition (24.28 cm) when K was top dressed (48.86 cm). In the case of pod production, BU Soybean-1 produced a numerically higher number of pods under control (18.86 and 18.53 pods plant^−1^ with basal and top dressing of K application, respectively) than WL condition (Table 2). The longest pod (4.31 cm) was measured in BU Soybean-2 under control when K was top dressed.

### 3.2. Production Seeds Plant^−1^ and 100-Seed Weight

Although seeds pod^−1^ did not vary significantly due to interaction of variety, K and WL, BU Soybean-2 produced a higher number of seeds pod^−1^ when K was applied basally under control (2.80 pod^−1^) followed by WL (2.60 pod^−1^) condition (Table 3). In the case of BU Soybean-2, the seeds plant^−1^ was 42.26 and 40.50 with the basal application of K in control and WL condition, respectively. The 100-seed weight of both genotypes was higher under the control condition compared to WL. However, the 100-seed weight of BU Soybean-2 was higher (22.04 g) when K was top dressed followed by basal application (19.73 g) in control.

### 3.3. Grain and Straw Yield of Soybean

The interaction of variety, K, and WL exhibited a significant effect on the grain and straw yield of soybean (Figure 1). BU Soybean-2 produced the highest grain yield compared to BU Soybean-1 in all growing conditions and modes of K application. This variety produced 2.63 and 2.84 t ha^−1^ grain in control and 1.65 and 1.71 t ha^−1^ under WL condition with basal and top dressing of K, respectively. BU Soybean-1 produced 1.33 and 1.36 t ha^−1^ under control and 1.24 and 1.23 t ha^−1^ under WL conditions when K was applied basally and top dressing, respectively (Figure 1). BU Soybean-2 produced a higher amount of straw yield compared to BU Soybean-1, and the straw yield of both genotypes was higher under control than WL condition (Figure 1). The straw yield of BU Soybean-2 was the highest (3.28 t ha^−1^) under control conditions when K was top dressed followed by basal application (3.15 t ha^−1^). Similar to grain yield, BU Soybean-1 gave higher straw yield under control than WL condition. BU Soybean-1 produced 1.66 and 1.63 t ha^−1^ under control and 1.66 and 1.45 t ha^−1^ under WL conditions when K was applied basally and top dressing, respectively (Figure 1). The HI of BU Soybean-2 was higher (0.46) under control conditions when K was top dressed. The lower (0.42) was obtained from WL control when K was applied basally in BU Soybean-1.

### 3.4. Nutrient Accumulation in Soybean Seed

The interaction of variety, K, and WL exhibited a significant effect on the N, P, and K content of soybean grains. Both varieties absorbed a higher amount of N under control conditions compared to WL (Figure 2). Between two soybean varieties, BU Soybean-2 accumulated a higher amount of N in grain. BU Soybean-2 absorbed the highest amount of N (188.04 kg ha^−1^) when K was top dressed followed by basal application under control conditions (170.93 kg ha^−1^). This variety accumulated 90.73 and 80.88 kg N ha^−1^ in grain under WL conditions when K was applied basally and top dressing, respectively. The N content of the grains of BU Soybean-1 was accumulated at 67.73 and 72.30 kg ha^−1^ in control and 60.35 and 59.09 kg N ha^−1^ in WL condition with basal and top dressing of K fertilizer, respectively.

Similar to N accumulation, a higher amount of P was taken up by BU Soybean-2 under control than WL condition (Figure 2). The P absorption in the grain of BU Soybean-2 was the highest (13.89 kg ha^−1^) under control conditions when K was applied basally followed by top dressing (12.90 kg ha^−1^). Again, BU Soybean-2 accumulated 7.28 and 7.00 kg P ha^−1^ in basal and top dressing of K, respectively, under WL conditions. In the case of BU Soybean-1, a higher amount of P was taken up under top dressed treatment in both growing conditions. BU Soybean-1 absorbed 7.73 and 7.20 kg P ha^−1^ under top dressing and 6.88 and 6.10 kg P ha^−1^ under basal application of K fertilizer in control and WL conditions, respectively (Figure 2).

Both varieties absorbed a higher amount of K under control conditions compared to WL. Between two soybean varieties, BU Soybean-2 accumulated a higher amount of K in grain. BU Soybean-2 absorbed the highest amount of K (77.89 kg ha^−1^) when K was top dressed followed by basal application under control conditions (74.52 kg ha^−1^). This variety accumulated 45.65 and 44.69 kg K ha^−1^ in grain under WL conditions when K was applied basally and top dressing, respectively. The K content of the grains of BU Soybean-1 was 42.96 and 42.77 kg ha^−1^ in control and 34.07 and 37.75 kg K ha^−1^ in WL condition with basal and top dressing of K fertilizer, respectively.

### 3.5. Protein Content and EC of Soybean Seed

The interaction of soybean, K, and WL exhibited a significant effect on the protein percentage of soybean (Table 4). Generally, BU Soybean-2 contained a higher amount of protein as compared to BU Soybean-1. Similarly, both soybean varieties produced higher amounts of protein under control conditions with top dress K application. The lowest amount of protein (28.48%) was found in BU Soybean-1 under WL condition when K was top dressed. BU Soybean-2 gave the highest amount of EC (129 μS cm^−1^ g^−1^) under WL condition when K was applied basally. The second highest EC (125 μS cm^−1^ g^−1^) was also obtained from BU Soybean-2. The lowest value of EC (82 μS cm^−1^ g^−1^) was found in BU Soybean-1 under control when K was top dressed (Table 4). Between two varieties, BU Soybean-2 produced the heaviest seed. Under control conditions, both varieties produced the highest amount of seed weight compared to WL. BU Soybean-2 produced the heaviest seed (220.40 mg seed^−1^) under control conditions when K was top dressed and the lowest (138.16 mg seed^−1^) in WL when K was applied basally (Table 4). In the case of BU Soybean-1, the heaviest (118.13 mg seed^−1^) seed was observed under control conditions, and the lighter one (88.86 mg seed^−1^) in WL condition when K was applied basally.

### 3.6. Germination and Seed Vigor Index of Soybean

The interaction of variety, K, and WL exhibited a significant effect on the germination index of soybean (Figure 3). Between the two varieties, BU Soybean-2 gave the highest germination index compared to BU Soybean-1. The germination index of BU Soybean-2 was higher (34.14) under the WL condition, while it was lower (29.56) in control when K was applied basally. In the case of BU Soybean-1, the germination index was 7.63 and 16.35 in the basal application and 16.35 and 12.50 in top dressed treatment under control and WL, respectively (Figure 3). In BU Soybean-2, the highest percentage of germination (80.0%) was found under the flooded condition when K was top dressed and a lower percentage (62.0%) in control when K was applied basally. In the case of BU Soybean-1, a higher germination percentage (34.00%) of germination was found in WL, and the lower one (14.66%) was observed in control when K was applied basally. BU Soybean-2 gave higher seed vigor index compared to BU Soybean-1. BU Soybean-2 gave the highest seed vigor index (8.45) under WL condition when K was top dressed, while the lowest one was observed in BU Soybean-1 in control when K was applied basally.

Between the two varieties, MGT was higher in BU Soybean-2 compared to BU Soybean-1. The MGT of BU Soybean-2 was 48.86 days under WL condition when K was top dressed followed by basal application. Similarly, MGT was 40.26 days in top dress treatment and 34.56 days for basal application of K in the control plot. However, MGT for BU Soybean-1 was 24.66 days when K was top dressed, while it was 8.60 days in basal K application plot under control conditions. Moreover, this variety needed 20.53 and 16.50 days for MGT in basal and top dress plots, respectively, under flooded conditions (Figure 4).

## 4. Discussion

Nutrient element K has a viable function in the development, growth, and production process of plants [43,55]. In the present experiment, the plant height of BU Soybean-1 decreased from 25.20 cm in control to 24.78 cm in WL condition when K was applied basally. However, taller plants were measured in both testing soybean materials when K was top dressed. Moreover, BU Soybean-2 produced taller plants than BU Soybean-1 under both growing conditions and modes of K applications. The genetic difference was responsible for the different plant heights of the two varieties. The WL-induced decrease in plant height was noted in soybean [56]. Jin-Woong et al. [57] reported that the reduction in plant height under WL conditions was probably due to oxygen deficiency, anaerobic conditions, less root activity, and inhibition of synthesis and transport of photosynthetic assimilates.

Production of pods plant^−1^ is an important yield-contributing characteristic. In this study, the number of pods plant^−1^ decreased due to the imposition of WL treatment. However, BU Soybean-1 produced a higher number of pods than BU Soybean-2 under control conditions. The WL induced several physiological disturbances in growth and pod formation [17]. Jin-Woong et al. [57] and Sathi et al. [58] found that the number of pod plant^−1^ was sharply reduced due to the imposition of WL. In this study, the application of K after the recession of flood water provokes the production of more pods plant^−1^ (Table 2). The supplementation of K increased photosynthetic capacity and Chl content reported, resulting in taller plants and maximum pods plant^−1^ [59,60]. On the other hand, the longer-sized pod was produced by BU Soybean-2 more than BU Soybean-1 under both growing conditions and mode of K application. This indicated that BU Soybean-2 can produce pods of longer length with a fewer number of pods plant^−1^. However, the lower number of pods plant^−1^ under WL conditions resulted in a lower yield [61,62,63]. The better field performance under WL conditions in terms of pod production with the split application of K after the recession of flood was also supported by previous findings [64,65]. 

The production of seed plant^−1^ or pod^−1^ and individual seed weight is directly related to grain yield. BU soybean-1 produced a higher number of seeds plant^−1^ under control compared to WL. However, both soybean varieties performed better for seed production under control with basal K application. Many experiments have explored the influence of the basal application of K on the yield and quality of wheat [66,67]. Comparatively, smaller seeds were found under WL conditions in both varieties. Ara et al. [37] and Sathi et al. [58] found that when the plants were subjected to WL stress, 100-seed weight decreased in comparison to the control condition. However, BU Soybean-2 produced a lower number of seeds plant^−1^ but gave bigger-sized seeds. Seed weight is a genetic characteristic, and BU Soybean-2 is a bold grain soybean variety. However, split application of K fertilizer increased the 100-grain weight of soybean varieties under WL conditions (Table 3). 

This indicated that the split application of K improved the production of seed through the use of another nutrient element by soybean plants. Ahmed et al. [68] reported that the test weight of maize and soybean increased by 8 and 4%, respectively, due to a higher amount of K application. The application of K at a higher rate increased photosynthesis and accumulation of a greater amount of photosynthate to grain [65,66,67,68,69], as split application of K after recession of flood water favors roots to absorb more minerals from the soil. On the other hand, K also helps to increase photosynthesis and production of more photo-assimilates that are ultimately stored in the seed. Thus, the 100-seed weight of BU Soybean-2 increased when K was applied after removal of flood.

The yield of soybean reduced significantly under WL conditions. The reduction of yield-contributing characteristics under WL (Table 2 and Table 3) resulted in lower yield (Figure 1). Islam et al. [70] also found that the number of pods plant^−1^, seed weight, and seed yield in mungbean were significantly affected due to soil WL stress. Amin et al. [39] and Vineela [63] recorded a significant decrease in seed yield in mungbean due to WL. In this experiment, BU Soybean-2 produced 3.6% more yield under WL conditions when K was applied after the recession of flood water as compared to the basal application of K fertilizer (Figure 1). This indicated that K fertilizer can reduce the detrimental effect of flooding. Vyas et al. [60] reported that K application significantly improved the seed yield of soybean. Uddin et al. [71] found that test weight and seed increased by K application. The greater yield and high-quality grains obtained due to K application might be due to increased photosynthesis, greater carbohydrate translocation toward the sink, and metabolism [44,72,73]. 

Similar to yield, the straw yield of BU Soybean-2 also increased by split application of K fertilizer under control as well as WL condition (Figure 1). This variety produced 1.92 and 1.95 t ha^−1^ straw in basal and top dressing of K, respectively, under WL conditions. A similar finding was also reported by Farhad et al. [74]. However, the detrimental effect of WL on the grain yield of soybean was also found by Beutler et al. [75] and Koger et al. [76]. Although there was no significant change in HI found in this study, Youn et al. [77] reported that HI increased only in WL soybeans. On the contrary, a reduction of HI due to WL was reported in mungbean [63,78]. According to Nguyen et al. [79], WL stress during the vegetative stage of soybean growth causes a reduction in grain yield of approximately 17–40%, and that the reproductive stage led to a 40–57% yield reduction. A strong positive relationship between K fertilizer input and grain yield has been shown [80].

Nutrient accumulation in seeds was more when K was applied after the recession of flood water (Figure 2). Ahmed et al. [68] found that all K applications improved the total N accumulation in plants. This helped to release ammonium ions from the soil and made N more available to plants. Increasing the K application increased the N, P, and K content of plants [81]. Board et al. [82] reported that the effects of WL on P were minor. It is now well established that metabolic energy is required for the active transport of N, P, and K through the root system [83,84]. Under hypoxic conditions, the stored metabolic energy of root cells is appreciably reduced, thereby suppressing the active transport of these nutrients [85,86].

Both testing varieties absorbed a higher amount of N, P, and K under control conditions compared to flooding. WL inhibits the uptake of most essential nutrients in the soil and thus leads to deficiencies in N, P, K, Mg, and Ca [87]. The N contents decreased markedly in different parts of the cotton plant under waterlogged conditions and exogenously applied K showed considerable improvement in N contents in all plant parts [88]. The application of K under WL conditions improved the accumulation of other plant nutrients such as K^+^, Ca^2+^, N, Mn^2+^, and Fe^2+^ [59]. 

BU Soybean-2 accumulated the highest amount of protein under control conditions with a top dressing of K fertilizer. The second highest protein content was also obtained from BU Soybean-2 under control conditions with basal K application (Table 4). Vyas et al. [60] found that protein content was significantly higher with a split application of 37.5 kg as basal + 37.5 kg K_2_O ha^−1^ at the flowering stage of the crop. Alam et al. [65] reported that the grain protein content of wheat was significantly influenced due to the application of different levels of K. During the seed quality test, BU Soybean-2 gave a higher EC value compared to BU Soybean-1. The higher EC value indicated the higher injury of the seeds during flooding. Wuebker et al. [89] observed that seeds absorbed water (imbibition) and reduced germination under WL conditions. Several reports showed a negative correlation between germination percentage and WL stress [90,91]. The germination test showed that exogenous application of K favored seed germination (Figure 3 and Figure 4) when the field was waterlogged for up to 4 days. Seed lots with a greater germination index is considered to be more vigorous [52]. The survival rate and germination percent were quickly lost due to the WL condition where the amount of oxygen was very low [92]. Potassium is a good catalyst for seed germination and emergence. Potassium nitrate, potassium chloride, and dipotassium hydrogen phosphate are the common K salts used in seed priming [83].

Heavy rainfall, high water tables and poor drainage create water logging in areas of the world [93]. Low-oxygen in soil under waterlogged conditions limits yield of soybean [94]. Waterlogging impacts around 10–12% of agricultural soils [95], and about 6 million tons of grain per year are lost due to this stress with economic losses of approximately US $ 1.5 billion annually [96].

In Bangladesh, the cropping intensity is high during the rabi (November to March) season, which is the best time for soybean cultivation. Different natural calamities start during the month of March, when the soybean crops attain pod formation to the maturity stage. In 2017, heavy rainfall occurred during the month of April, and the torrential rains damaged the soybean at the pod development stage in the Noakhali and Bhola districts of Bangladesh. Therefore, the application of K after flooding will reduce damage to soybean and increase production, improve farmer income and ensure national food security in Bangladesh. However, the intensity of flooding damage caused during the vegetative stage and its impact on seed quality should be addressed in future research.

## 5. Conclusions

Waterlogging showed a detrimental effect on pods and seeds plant^−1^, pod length, 100-seed weight, grain and straw yield, nutrient, and protein accumulation in soybean grain. On the other hand, flood-affected seeds had higher germination percent, seed vigor index, and electrical conductivity, and needed more mean germination time for both soybean varieties. Basal application of potassium fertilizer improved height of plant, pods, and seeds plant^−1^, gave higher electrical conductivity and needed more mean germination time of both soybean varieties. On the contrary, top dressing (50% as basal + 50% as top dress after the termination of flooding) increased 100-seed weight, grain and straw yield, nutrient, protein accumulation in grain, germination percent, and seed vigor index of soybean under both control and flooding. Therefore, it might be concluded that exogenous application of K fertilizer after the recession of flood water could be recommended for higher grain yield in flood-affected soybean growing areas.

## Figures and Tables

**Figure 1 life-12-01816-f001:**
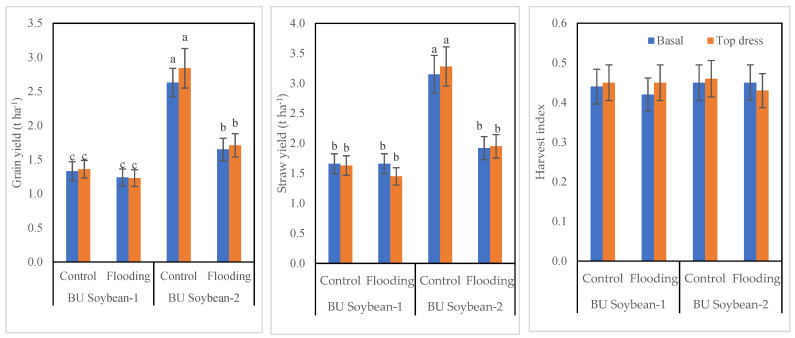
Interaction effect of variety, K and WL on grain and straw yield of soybean. Bar graphs indicate mean value ± standard error. Bars with similar letters did not differ significantly at *p* < 0.05 level.

**Figure 2 life-12-01816-f002:**
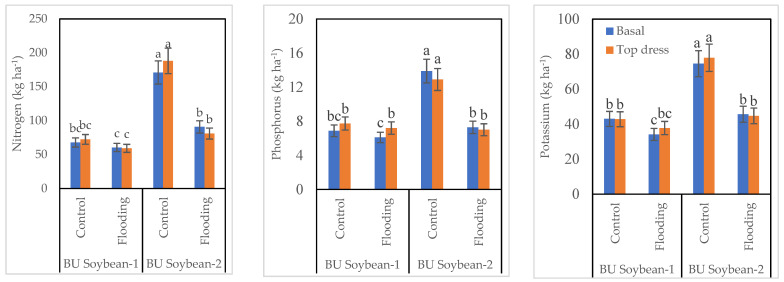
Interaction effect of variety, K and WL on nutrient contents in soybean seeds, Bar graphs indicate mean value ± standard error. Bars with similar letters did not differ significantly at *p* < 0.05 level.

**Figure 3 life-12-01816-f003:**
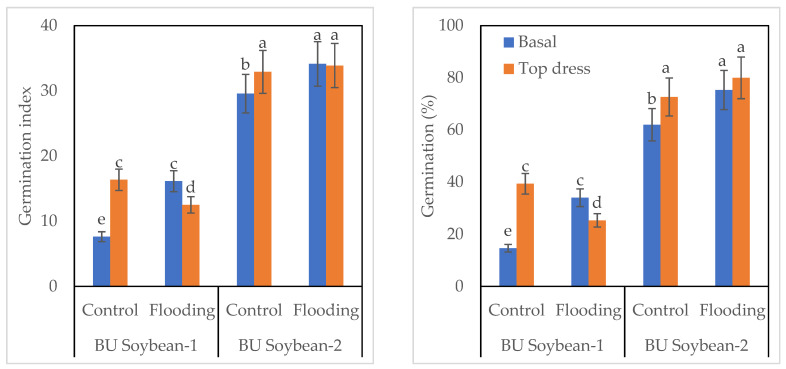
Effect of variety, K and WL on germination of soybean seeds. Bar graphs indicate mean value ± standard error. Bars with similar letters did not differ significantly at *p* < 0.05 level.

**Figure 4 life-12-01816-f004:**
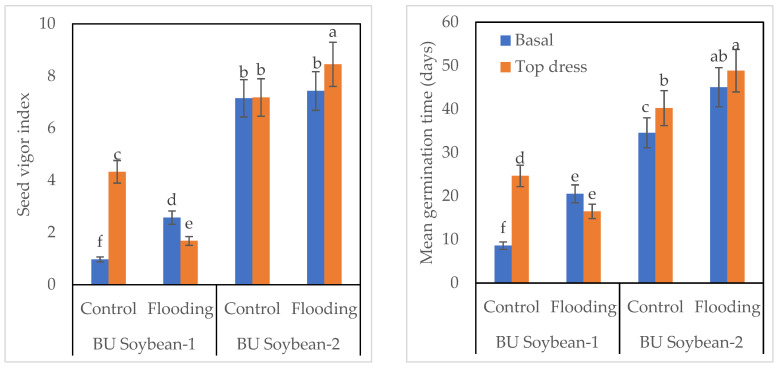
Effect of variety, fertilizer, and WL on vigor index and mean germination time of soybean seeds. Bar graphs indicate mean value ± standard error. Bars with similar letters did not differ significantly at *p* < 0.05 level.

**Table 1 life-12-01816-t001:** Temperature, precipitation, and relative humidity of the research site during the experiment period.

Months and Metrological Events	January	February	March	April	May	June
Average temperature (°C)	17.8	20.0	25.2	28.3	29.0	30.2
Maximum temperature (°C)	28.8	26.5	32.3	34.7	33.4	33.4
Minimum temperature (°C)	12.8	13.5	18.5	21.8	24.8	26.7
Relative humidity (%)	87	73	80	82	84	84.5
Total precipitation (mm)	31.8	2.3	16	40.2	290.5	416.3

**Table 2 life-12-01816-t002:** Effect of variety, fertilizer, and WL on plant height and pod production of soybean.

Soybean Varieties	Growing Condition	Plant Height (cm)		Pods Plant^−1^		Pod Length (cm)	
		Basal K	Top Dress K	Basal K	Top Dress K	Basal K	Top Dress K
BU Soybean-1	Control	25.20 c	24.28 c	18.86	18.53	3.98 ab	3.56 c
	WL	24.78 c	24.48 c	14.40	14.73	3.76 b	3.63 c
BU Soybean-2	Control	42.83 b	41.20 b	14.46	15.66	4.31 a	3.78 b
	WL	48.14 a	48.86 a	14.93	16.20	4.16 a	4.18 a

WL, water logging, WL, water logging, Figures with similar letters in a column did not vary significantly.

**Table 3 life-12-01816-t003:** Effect of variety, fertilizer, and WL on seed production and 100-seed weight of soybean.

Soybean Varieties	WL	Seeds Pod^−1^		Seeds Plant^−1^		100-Seed Weight (g)	
		Basal K	Top Dress K	Basal K	Top Dress K	Basal K	Top Dress K
BU Soybean-1	Control	2.66	2.45	50.54 a	45.48 ab	11.81 c	11.10 c
	WL	2.33	2.00	32.98 c	30.40 c	8.88 d	9.75 d
BU Soybean-2	Control	2.80	2.13	42.26 ab	39.08 b	19.73 a	22.04 a
	WL	2.60	2.60	40.50 ab	33.09 c	13.81 c	16.44 b

WL, water logging, WL, water logging, Figures with similar letters in a column did not vary significantly.

**Table 4 life-12-01816-t004:** Effect of variety, fertilizer, and WL on protein, EC, and weight of soybean seed.

Soybean Varieties	Flooding	Protein (%)		EC (μS cm^−1^ g^−1^)		Seed Weight (mg seed^−1^)	
		Basal K	Top Dress K	Basal K	Top Dress K	Basal K	Top Dress K
BU Soybean-1	Control	30.19 b	31.48 b	108 a	82 c	118.13 b	111.06 b
	WL	28.85 c	28.48 c	89 c	92 b	88.86 c	97.50 c
BU Soybean-2	Control	38.60 a	39.35 a	106 b	92 b	197.33 a	220.40 a
	WL	32.61 b	31.83 b	129 a	125 a	138.16 ab	164.46 ab

WL, water logging, WL, water logging, Figures with similar letters in a column did not vary significantly.

## Data Availability

Data recorded in the current study are available in all tables and figures of the manuscript.

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
