# Peer review of "Application of Potassium after Waterlogging Improves Quality and Productivity of Soybean Seeds"

_life, 2022, doi:10.3390/life12111816_

Round 1

Reviewer 1 Report

This work investigated the effects of applying potassium in waterlogged soybean. The topic is interesting and well within the aims of the Journal, but it needs minor revisions before it can be published.   Here some tips to improve the work:   TITLE I suggest to simplify and shorten the title, such as the following: “Application of potassium after waterlogging improves quality and productivity of soybean seeds”.   KEYWORDS Why do you choose “chlorofyll”? I don't think it is a keyword of the work.   INTRODUCTION Lines 48-51 I suggest implementing the concept by specifying the specific nutraceutical characteristics of soybean. Line 71, what do you mean with “and detoxify the ROS”? Please, explain better. Lines 84-90 references are missing.   MATERIALS AND METHODS I suggest moving the lines 121-123 in line 113 before “the air temperature…” Line 135, what do you mean with “blocks”? how did you calculate the moisture of the seeds? It is not clear paragraph 2.8 make uniform the line spacing please. Lines 209-213 change all “is” in “was” please. Line 214, I suppose that N and D should have been written in lowercase like the caption or vice versa

Line 215, all the sentence should be move into the discussion chapter. 

Lines 223 and 224 it is a repetition of what was written just before.

Figure 2, delete the picture and caption box.

Line 335, insert at the end “(Figure 4)” 

Author Response

Reviewer 1 (round 1)

Comments: This work investigated the effects of applying potassium in waterlogged soybean. The topic is interesting and well within the aims of the Journal, but it needs minor revisions before it can be published.  

Author response: Thank you for allowing us the opportunity to submit our revised manuscript for publication in the Life Journal of MDPI. We appreciate the time and effort you have taken to improve our manuscript. We are also thankful to the honorable reviewer for the positive decision to publish. We revised our manuscript following your point-by-point comments and suggestions for substantial improvement. We hope that this revised version satisfies you to take the final decision.

Comments: Here some tips to improve the work:   TITLE I suggest to simplify and shorten the title, such as the following: “Application of potassium after waterlogging improves quality and productivity of soybean seeds”.  

Author response: Thank you for your comments. We have changed the title according to your suggestion.

 Comments: KEYWORDS Why do you choose “chlorophyll”? I don't think it is a keyword of the work.  

Author response: Thank you for your comments. We have removed it.

Comments: INTRODUCTION Lines 48-51 I suggest implementing the concept by specifying the specific nutraceutical characteristics of soybean.

Author response: Thank you for your comments. The 100 g dry seeds of soybean contain 30 – 50% of protein, 277 mg calcium, 15.7 mg iron, 280 mg magnesium, 704 mg phosphorus, 1797 mg potassium and 375 µg folic acid.

Comments: Line 71, what do you mean with “and detoxify the ROS”? Please, explain better.

Author response: Thank you for your comments. We have replaced the word “and” with “which”.

For explanation, this is well established in literature that to cope with drought plants enhanced both enzymatic and non-enzymatic antioxidants. The enzymatic and non-enzymatic antioxidants detoxify the ROS produced due to drought stress effects. This is the self-mechanism of mitigation of drought stress in plants, especially in tolerant plants.

Comments: Lines 84-90 references are missing.  

Author response: Thank you for your comments. We have a single reference 37 for these sentences.

Comments: MATERIALS AND METHODS I suggest moving the lines 121-123 in line 113 before “the air temperature…”

Author response: Thank you for your comments. We have moved the sentences.

 Comments: Line 135, what do you mean with “blocks”? how did you calculate the moisture of the seeds? It is not clear paragraph 2.8 make uniform the line spacing please.

Author response: Thank you for your comments. The blocks mean replication/repetition of treatments. Each block indicates a replication. The seed moisture content was determined with a digital moisture meter.  

Comments: Lines 209-213 change all “is” in “was” please.

Author response: Thank you for your comments. We have changed “is” in “was”.

 Comments: Line 214, I suppose that N and D should have been written in lowercase like the caption or vice versa

Author response: Thank you for your comments. We have revised it.

 Comments: Line 215, all the sentence should be move into the discussion chapter. 

Author response: Thank you for your comments. We have moved the sentence from materials and methods to discussion.  

 Comments: Lines 223 and 224 it is a repetition of what was written just before.

Author response: Thank you for your comments. We have removed the repetition.

 Comments: Figure 2, delete the picture and caption box.

Author response: Thank you for your comments. We have removed the box.

 Comments: Line 335, insert at the end “(Figure 4)” 

Author response: Thank you for your comments. We have added Table 4.

Reviewer 2 Report

Comments to the authors:

Authors analyzed the effect of potassium and waterlog on different soybeans. This is an interesting study, which provide new cues to further uncover the mechanism of soybean waterlogging-enduring. The MS should be published after revised.

All my questions about this article are as follows:

1.       Although the official language of Bangladesh includes English. However, there are numerous other smaller issues, but it would be easier to address these once this paper is in better shape.

2.       Authors should add more recent references in introduction and discussion section, about 3-5 years.

3.       Abstract section, I suggest that the author add a description about soybean at the beginning, because the research object of this paper is soybean.

4.       Introduction section, Line47, authors should update the data on soybean acreage.

5.       Table 3, Top dress K could increase the 100-seed weight in BU soybean-2, how do authors explain this situation?

6.       Table 4, the description of the processing is different from the other tables, Basal and Basal K? Top dress and Top dress K? Please modify it.

7.       Discussion section, authors should add more practical application of this topic and future work to be conducted in soybean production of Bangladesh.

8.       As we all know, Bangladesh is a country with many floods. The author's research is of great practical significance. I hope the author's achievements can be applied to soybean agricultural production as soon as possible.

Author Response

Reviewer 2 (round 1)

Comments: Authors analyzed the effect of potassium and waterlog on different soybeans. This is an interesting study, which provide new cues to further uncover the mechanism of soybean waterlogging-enduring. The MS should be published after revised.

Author response: Thank you for allowing us the opportunity to submit our revised manuscript for publication in the Life Journal of MDPI. We appreciate the time and effort you have taken to improve our manuscript. We are also thankful to the honorable reviewer for the positive decision to publish. We revised our manuscript following your point-by-point comments and suggestions for substantial improvement. We hope that this revised version satisfies you to take the final decision.

Comments: All my questions about this article are as follows:

Comments: 1. Although the official language of Bangladesh includes English. However, there are numerous other smaller issues, but it would be easier to address these once this paper is in better shape.

Author response: Thank you for your comments.

Comments: 2. Authors should add more recent references in introduction and discussion section, about 3-5 years.

Author response: Thank you for your comments. We have added most recent references in introduction and discussion section

Comments: 3.  Abstract section, I suggest that the author add a description about soybean at the beginning, because the research object of this paper is soybean.

Author response: Thank you for your comments. We have added the sentences “Soybean is a promising crop that can easily fit with the cropping pattern during kharif I season, when water logging occurs due to sudden rain.”  

Comments: 4. Introduction section, Line47, authors should update the data on soybean acreage.

Author response: Thank you for your comments. We have updated the acreage

Comments: 5. Table 3, Top dress K could increase the 100-seed weight in BU soybean-2, how do authors explain this situation?

Author response: Thank you for your comments. We have added in the discussion “As split application of K after recession of flood water, favors root to absorb more minerals from soil. On the other hand, K also helps to increase photosynthesis and production of more photo assimilates that ultimately stores in the seed. Thus, the 100-seed weight of BU Soybean-2 increased when K applied after removal of flood.”

 Comments: 6. Table 4, the description of the processing is different from the other tables, Basal and Basal K? Top dress and Top dress K? Please modify it.

Author response: Thank you for your comments. We have corrected these as Basal K and Top dress K.

Comments: 7.  Discussion section, authors should add more practical application of this topic and future work to be conducted in soybean production of Bangladesh.

Author response: Thank you for your comments. We have added some discussion.

Comments: 8.  As we all know; Bangladesh is a country with many floods. The author's research is of great practical significance. I hope the author's achievements can be applied to soybean agricultural production as soon as possible.

Author response: Thank you for your comments. We also hope that the findings will help to reduce the flood effect in case of soybean cultivation.  
